# Our Mental Health Is Determined by an Intrinsic Interplay between the Central Nervous System, Enteric Nerves, and Gut Microbiota

**DOI:** 10.3390/ijms25010038

**Published:** 2023-12-19

**Authors:** Leon M. T. Dicks

**Affiliations:** Department of Microbiology, Stellenbosch University, Private Bag X1, Matieland, Stellenbosch 7602, South Africa; lmtd@sun.ac.za

**Keywords:** gut microbiota, enteric nervous system, mental health

## Abstract

Bacteria in the gut microbiome play an intrinsic part in immune activation, intestinal permeability, enteric reflex, and entero-endocrine signaling. The gut microbiota communicates with the central nervous system (CNS) through the production of bile acids, short-chain fatty acids (SCFAs), glutamate (Glu), γ-aminobutyric acid (GABA), dopamine (DA), norepinephrine (NE), serotonin (5-HT), and histamine. A vast number of signals generated in the gastrointestinal tract (GIT) reach the brain via afferent fibers of the vagus nerve (VN). Signals from the CNS are returned to entero-epithelial cells (EES) via efferent VN fibers and communicate with 100 to 500 million neurons in the submucosa and myenteric plexus of the gut wall, which is referred to as the enteric nervous system (ENS). Intercommunications between the gut and CNS regulate mood, cognitive behavior, and neuropsychiatric disorders such as autism, depression, and schizophrenia. The modulation, development, and renewal of nerves in the ENS and changes in the gut microbiome alter the synthesis and degradation of neurotransmitters, ultimately influencing our mental health. The more we decipher the gut microbiome and understand its effect on neurotransmission, the closer we may get to developing novel therapeutic and psychobiotic compounds to improve cognitive functions and prevent mental disorders. In this review, the intricate control of entero-endocrine signaling and immune responses that keep the gut microbiome in a balanced state, and the influence that changing gut bacteria have on neuropsychiatric disorders, are discussed.

## 1. Introduction

Most neurological signals to and from the gut run through a bidirectional vagus nerve (VN) that exits the brain at the medulla oblongata. Signals from the central nervous system (CNS), passed along efferent VN fibers, reach 100 to 500 million neurons in the ENS [1] that surround the gastrointestinal tract (GIT). The enteric nervous system (ENS), often referred to as the “second brain”, is by far the largest of all the nerve systems in the human body and functions independently from the VN [2]. Experiments on adult mice have shown that myenteric neurons have a short life span and are subjected to infrequent episodes of apoptosis [3]. According to Kulkarni et al. [4], more than 85% of myenteric neurons in the small intestine of mice are replaced within two weeks but are replaced at a constant rate. Newly formed neurons within myenteric ganglia express the neuroepithelial stem cell protein nestin, the nuclear protein Ki67, and the neurotrophin receptor p75^NTR^ [4]. Nestin maintains the balance between neuronal apoptosis and neurogenesis [4,5], Ki67 is associated with rRNA transcription, and p75^NTR^ decreases the activation of Rho, thus favoring axonal elongation [6]. A decline in the health of neural stem cells (NSCs) will ultimately lead to a deficiency in the production of any of these proteins and the inability to renew damaged cells [7].

To fully understand the interaction between the GIT and the brain and how the gut–brain axis functions, it is important to understand the role of intrinsic and extrinsic sensory neurons in the ENS. Neurons in the ENS are interconnected via internodal strands with axons up to 13 cm in length to form two distinct ganglionated neuronal plexuses, referred to as the myenteric and submucosal plexuses [8]. Smooth muscle cells of the GIT are in close contact with excitatory and inhibitory motor neurons [9,10,11]. Motor neurons in the myenteric plexus coordinate muscle movements (peristalsis), while neurons in the submucosal plexus regulate secretion and absorption [8]. The major extrinsic neural pathways between the ENS and the CNS are shown in Figure 1. Vagal dorsomotor efferent fibers connect with parasympathetic enteric nerves of the stomach and upper part of the small intestine. Enteric nerves in the rest of the small intestine are connected to sympathetic nerves in the coeliac and superior mesenteric ganglia. Enteric nerves of the large intestine and distal part of the colon are connected to sympathetic efferent nerves in the inferior mesenteric ganglia and parasympathetic nerves in the pelvic ganglia. Motor and sensory neurons in the submucosa and myenteric plexus of the gut wall regulate muscle activity, gut wall motility, secretion of fluids, and blood flow [12]. The large intestine, specifically the distal colon, has afferent nerves linked to the spinal cord [13]. These signals reach enterochromaffin cells (ECs) and enteroendocrine cells (EECs) that are in direct contact with an estimated 4 trillion gut microorganisms. The celiac branch of the VN connects with the duodenum and the rest of the intestine to the distal part of the descending colon [14]. This intricate connection of neurons facilitates a rapid transfer of signals throughout the GIT [15,16]. The modulation, development, and renewal of neurons of the ENS are controlled by gut microbiota, especially those with the ability to produce and metabolize hormones (reviewed by Dicks [17]).

Dogiel type I neurons (Figure 2) are in the myenteric plexus of the human colon [18] and act as interneurons or motor neurons. Based on experiments conducted on the colon of guinea pigs, Dogiel type I neurons are short and are characterized by having one axon and up to twenty dendrites with broad or lamellar endings [19]. Dogiel type I neurons react to sensory stimuli from the GIT but also synaptic signals received from other neurons [20,21]. Dogiel type II neurons (Figure 2) have a single axon with up to 16 long dendrites and react to chemical and mechanical stimuli, rendering them sensitive to changes in muscle tension [8,22]. Dogiel type II neurons may react on the release of substances such as serotonin from enterochromaffin cells (ECs), but there is no direct evidence supporting this [23].

At least 10 different types of EECs have been characterized. Apart from regulating the secretion of serotonin, they are involved in the secretion of chromogranin/secretogranin, neuropeptide Y (NPY), vasoactive intestinal peptide (VIP), cholecystokinin (CCK), somatostatin, glucagon-like peptide (GLP)-1/2, ghrelin, and substance P (SP) [24,25]. Receptors on these sensory cells are expressed by gut enteric neurons but also vagal afferents, the brainstem, and the hypothalamus [26,27]. For more information on sensorial signals, neural circuits, and gut peristalsis regulated by the ENS, the reader is referred to the review by Spencer and Hu [28].

Signals generated by gut microbiota, either in response to signals received from the CNS or other organs, or generated because of gastrointestinal metabolic reactions, are returned to the brain via afferent VN fibers [14]. Afferent fibers outnumber efferent fibers by 9:1, which indicates the importance of signaling from the GIT to the CNS. Molecules produced by gut microbiota regulate the CNS, entero-endocrine pathways, and the immune system. These regulatory or signaling molecules range from simple compounds such as bile acids, short-chain fatty acids (SCFAs), glutamate (Glu), aspartate, D-serine, and histamine to more complex structures such as γ-aminobutyric acid (GABA), dopamine (DA), norepinephrine (NE, also called noradrenaline, NAd), and serotonin (5-HT) [13,29,30,31]. For more information on neurotransmitters and the modulation, development, and renewal of ENS neurons, the reader is referred to the review by Dicks [17]. Communication amongst gut microbiota via quorum sensing (QS) signals and the impact of these signals on the CNS, including mental health, are reviewed by Dicks [32] and will be briefly discussed. The CNS is not discussed in detail.

## 2. Maintenance of the ENS

The ENS stems from enteric neural crest cells (ENCCs) [33]. The proliferation and migration of ENCCs are regulated by the glial-cell-line-derived nerve growth factor (GDNF) protein that is predominantly expressed by neurons in the septum, striatum, and thalamus [33]. The expression of GDNF and other neurotrophic factors, such as neurturin (NTN), artemin (ART), and persephin (PSP), is regulated by Toll-like receptors TLR2, TLR4, TLR5, and TLR9 [34]. GDNF, NTN, ART, and PSP bind to growth factor receptors (GFR)α-1, GFRα-2, GFR α3, and GFRα-4, respectively, all attached to the transmembrane receptor tyrosine kinase (RET) located on the cell membrane. The GFRα1–GDNF complex changes to the active tyrosine-phosphorylated form, which then signals to ENCCs to express ENS precursors and neurons [35].

Phosphatidylinositol 3-kinase (EC 2.7.1.137) and Akt (protein kinase B) in the PI3K/AKT signaling pathway stimulate neural survival and transmission. Phospholipase C gamma 1 (PLC-γ1; EC 3.1.4.3) is involved in cell growth, apoptosis, and transmission of neural cells. A ras/mitogen-activated protein kinase (RAS/MAPK) regulates neurogenesis [4,36]. ENS cells destroyed by apoptosis are replaced by newly formed cells [4]. However, little is known about the mechanisms that control ENS cell replenishment. Vicentini et al. [37] have shown that the small intestinal tract of antibiotic-induced mice (thus without gut microbiota) was longer than that observed in normal mice. This led to slower transit of gut contents, increased carbachol (carbamylcholine)-stimulated ion secretion, and an increase in gut wall permeability. Increased levels of carbachol activate acetylcholine (Ach) receptors [38], stimulating the formation of muscarinic and nicotinic receptors. Ach is the main neurotransmitter of the parasympathetic nervous system and plays a role in the contraction of smooth muscles, dilation of blood vessels, secretion of bodily fluids, and the downregulation of heart rate. Muscarinic Ach receptors form G-protein-coupled receptor complexes in the membranes of parasympathetic nerve cells [38]. Polypeptide nicotinic acetylcholine receptors are present in the central and peripheral nervous system, muscle, and many other tissues [39]. Vicentini et al. [37] also noted a decrease in neurons in the submucosal and myenteric plexuses of the ileum and proximal colon of germ-free mice. The myenteric plexus of the ileum contained fewer glial cells (neuroglia) [37], which is an indication that neurons of the ENS had been deprived of nutrient and oxygen supply. Neuronal cells that were not insulated from each other did not destroy pathogens and could not remove dead neurons [40].

Research conducted on experimental animals has shown that Ach is stored in vesicles at the terminals of Ach-producing neurons. The secretion of Ach is most likely stimulated by certain *Lactobacillus* spp. [41,42]. A decrease in the production of Ach was recorded when blood flow was restricted at the proximal cerebral artery, which altered the composition of the gut microbiota [43,44]. Furthermore, a decrease in the production of Ach resulted in irregular peristalsis, dysbiosis, an increase in gut permeability [43,44], and inflammation of the GIT [45]. This, in turn, restricted the middle cerebral artery, which led to a decline in goblet cells in the cecum and a lowering of mucin production [45]. The influence that a single neurotransmitter such as Ach has on the parasympathetic nervous system, and the gut microbiome, illustrates the sensitivity of ENS to changing conditions.

## 3. The Influence of Gut Bacteria on the ENS

The human gut is host to 2766 microbial species, of which more than 90% belong to the phyla Proteobacteria, Firmicutes, Actinobacteria, and Bacteroidetes [46,47,48,49,50,51,52]. The remaining 10% of the gut microbiome is composed of Fusobacteria and Verrucomicrobia [53]. Gut bacteria communicate with the CNS by using GABA, DA, NE, 5-HT, histamine [29], SCFAs [30], tryptophan [31], and secondary bile acids [13]. Signals generated by these neurotransmitters and molecules are transported to the CNS via afferent VN fibers. Signals from the brain reach ECCs and EECs in the gut wall and the mucosal immune system via efferent VN fibers [54]. This bidirectional communication improves the integrity of the gut wall, reduces peripheral inflammation, and inhibits the release of pro-inflammatory cytokines [55]. On another level, signals generated by the hypothalamus reach the pituitary and adrenal glands and communicate with EECs via the hypothalamic pituitary adrenal axis (HPA) [56]. The intricate control of entero-endocrine signaling and immune responses keeps the gut microbiome in a balanced state and prevents dysbiosis or the development of major GI disorders such as diarrhea, ulcerative colitis (UC), Crohn’s disease (CD), and inflammable bowel diseases (IBDs) [57,58,59]. Toxins produced by pathogenic bacteria may alter the functioning of the intestinal barrier and blood–brain barrier (BBB), and lead to neurodegeneration [60,61,62,63].

The importance of gut microbiota, lipopolysaccharides (LPS), and SCFAs in the recovery of enteric neurons was illustrated by experiments conducted on mice [37]. It is, however, important to note that LPS stimulated the recovery of damaged neurons and the proliferation of gut microbiota but not the formation of new neurons. SCFAs, on the other hand, restored neuronal loss [37]. Concluded from these results, a decrease in SCFAs, as expected in patients with dysbiosis, may lead to a loss of enteric neurons and a weakened ENS. This emphasizes the importance of SCFA-producing gut microbiota. SCFAs, such as butyrate, acetate, lactate, and propionate, are largely produced in the colon by *Bifodobacterium*, *Lactobacillus*, *Lachnospiraceae*, *Blautia*, *Coprococcus*, *Roseburia*, and *Faecalibacterium* [64]. SCFA receptors such as GPR43 (FFAR2) and GPR41 (FFAR3) are located on the surface of IECs [65] and are also expressed in the ENS, portal nerve, and sensory ganglia [5]. GPR 41 in the ENS transfers signals induced by SCFAs directly to the CNS [66]. GPR43 communicates with SCFAs to stimulate energy expenditure in skeletal muscles and the liver [67].

Glutamate (Glu) released from presynaptic nerve terminals adheres to ionotropic glutamate receptors (iGluRs) located on postsynaptic terminals. This increases the transfer of Ca^2+^ through voltage-activated Ca channels (VACCs) located in the terminals and activates calcium-calmodulin-dependent kinase (CaMK; EC 2.7.11.17), extracellular-signal-regulated kinase (ERK, EC 2.7.11.24), and the cyclic AMP response element binding (CREB) protein [68]. Excess Glu is converted to glutamine, which is transported back to presynaptic nerve terminals and converted to Glu by glutaminase (EC 3.5.1.2). The newly formed Glu is then transported to vesicles in the presynaptic neuron by vesicular glutamate transporters VGLUT1 and VGLUT2. This process is carefully controlled as Glu excitotoxicity accelerates the progression of Alzheimer’s disease [69]. L-Glutamate (L-Glu) is also converted to D-Glu by *Corynebacterium glutamicum*, *Lactobacillus plantarum*, *Lactococcus lactis*, *Lactobacillus paracasei*, *Brevibacterium avium*, *Mycobacterium smegmatis*, *Bacillus subtilis,* and *Brevibacterium lactofermentum* [49,70,71]. D-Glu is then decarboxylated by Glu decarboxylase (GAD; EC 4.1.1.15) to GABA [72,73]. The adhesion of GABA to GABA receptors (GABARs) on postsynaptic neurons prevents the transfer of Na^+^, K^+^, Ca^2+^, and Cl^−^ [74]. *Lactobacillus rhamnosus* JB-1 altered the expression of GABARs in the brain, which led to a decrease in anxiety and depression [75]. Treatment with *Lactobacillus* elevated GABA levels in the hippocampal and prefrontal cortex [76].

GABA, also present in millimolar concentrations in the brain, is released into the synaptic cleft upon depolarization of presynaptic neurons [77,78]. In these neurons, GABA α-oxoglutarate transaminase (GABA-T, EC 2.6.1.19) converts α-ketoglutarate to L-glutamic acid, which is then decarboxylated to GABA by GAD. Glial cells do not express GAD and cannot decarboxylate Glu to GABA. Glial cells convert GABA to succinic semialdehyde and glutamine, which is then deaminated to glutamate before it re-enters the GABA shunt [77].

Acetate produced by microbiota in the colon is transferred across the BBB and enters GABA neuroglial cycling pathways [79]. Gut-derived GABA reaches the CNS via specific GABA transporters expressed in the BBB, of which at least six have been identified [80]. These GABA transporters regulate the uptake of GABA into presynaptic nerve terminals and surrounding glial cells [81]. The reuptake of GABA by neurons results in decreased uptake of Na^+^. *Akkermansia muciniphila*, *Parabacteroides merdae*, and *Parabacteroides distasonis* alter GABA and glutamate ratios and increase glutamate levels in the brain [82].

Dopamine (DA) is produced in the substantia nigra, ventral tegmental area, and hypothalamus. Although DA is generally referred to as the reward neurotransmitter, it also plays a role in behavior and cognition, voluntary movement, motivation, inhibition of prolactin production, sleep patterns, mood, attention, working memory, and learning [83]. Tyrosine is hydroxylated to L-dihydroxyphenylalanine (L-DOPA) and then decarboxylated to DA. In the presence of β-hydroxylase, DA is converted to NE and Epi (Ad) [84]. DA is also produced by certain *Bacillus* and *Serratia* species in the GIT [85]. *Enterococcus faecalis* decarboxylates L-DOPA to DA but is then immediately dehydroxylated to m-tyramine by *Eggerthella lenta* [86].

Norepinephrine (NE) or Noradrenaline (NAd) is structurally similar to EPi (Ad). NE is produced during excitement but also plays a role in behavior, cognition [87], and inflammatory responses of the autonomic nervous system [88]. Low levels of NE (0.45 and 2.13 mM) are produced by *Bacillus mycoides*, *Bacillus subtilis*, *E. coli* K12, *Proteus vulgaris*, and *Serratia marcescens* [89]. An interesting finding was that NE stimulates the growth of *Klebsiella pneumoniae*, *Shigella sonnei*, *Pseudomonas aeruginosa*, *Enterobacter cloacae*, and *Staphylococcus aureus* [90].

Enteric serotonin levels are regulated by tryptophan hydroxylase TPH1 and serotonin produced from the ENS by tryptophan hydroxylase TPH2 [91]. Serotonin (5-HT) regulates appetite, gut motility, mood, cognition, and sleeping patterns [92,93,94]. As much as 80% 5-HT is produced in the GIT by *E. coli*, *Hafnia*, *Bacteroides*, *Streptococcus*, *Bifidobacterium*, *Lactococcus*, *Lactobacillus*, *Morganella*, *Klebsiella*, *Propionibacterium*, *Eubacterium*, *Roseburia*, and *Prevotella* [92,93]. Serotonin production by gut microbiota may have a greater effect on the CNS than originally anticipated as ECs interact with 5-HT-receptive afferent fibers in vagal or dorsal root neurons [95]. Physiological concentrations of SCFAs can cause an eight- to ten-fold increase in serotonin production, as shown in an in vitro colonic mucosal system [58]. The production of serotonin is carefully controlled by transporting excess concentrations across the cell membrane by a serotonin reuptake transporter (SERT), followed by the conversion to an inactive form by monoamine oxidase (MAO) [91]. This is important as high levels of serotonin may decrease the permeability of the gut wall [96]. Low levels of serotonin, on the other hand, lead to a decrease in the expression of occludin, which increases gut wall permeability. The latter was reported in patients diagnosed with IBS [96]. For further information on the effect of 5-HT on the ENS, the reader is referred to the review by Dicks [17].

Minor activation of the VN results in drastic changes in the production of neurotransmitters, which affects digestion, intestinal permeability, gastric motility, and immune regulation. This, in turn, alters the microbial composition in the GIT and may benefit the survival and proliferation of certain species. Neurohormones that may be affected by altering gut microbiota are oxytocin (Oxt) produced by the hypothalamus and vasopressin, also known as antidiuretic hormone (ADH) [97]. As soon as a neurotransmitter binds to the receptor on the postsynaptic membrane, the rate at which Ca^2+^, Na^+^, K^+,^, and Cl^−^ are transferred across ligand-gated channels changes, which results in either a stimulatory response or an inhibitory response [97]. These intricate and highly interweaved signaling mechanisms help to keep the gut microbiome in a balanced state. If unbalanced, the gut enters a state of dysbiosis, characterized by an increase in Enterobacteriaceae, especially *Escherichia*, *Shigella*, *Proteus*, and *Klebsiella* [60]. If left untreated, gastrointestinal disorders such as diarrhea, ulcerative colitis (UC), CD, and other inflammable bowel diseases (IBDs) may develop [57,58,59]. In severe cases, toxins produced by these bacteria may alter the functioning of the gut–blood barrier (GBB) and blood–brain barrier (BBB), ultimately leading to neurodegeneration [60,61,62,63].

## 4. Enteroendocrine Signaling and Immune Response

Enteroendocrine cells (EECs) comprise 1% of the human epithelium but resemble the largest endocrine system. These cells respond to luminal nutrients by secreting more than 20 peptide hormones, as stated elsewhere in this review. Most studies on EECs focused on their role in the postprandial assimilation of nutrients via endocrine- and paracrine-induced changes in gastrointestinal secretion, motility, the release of insulin, and satiety [98]. In the duodenum, GPRs of EECs sense long-chain fatty acids, triggering a Ca^2+^ flux, membrane depolarization, and secretion of CCK. CCK causes contraction of the gall bladder and the secretion of pancreatic enzymes, which in turn assimilate long-chain fatty acids [99].

Less well-studied are the response of EECs to pathogens (e.g., the expression of Toll-like receptors, TLRs) and metabolites produced by gut microbiota (referred to as the immunoendocrine axis). Immune cells have several receptors that recognize enteroendocrine-secreted hormone peptides [100], which support bidirectional signaling in the immunoendocrine axis. Peptide hormones and cytokines expressed by EECs react with receptors on innate and adaptive immune cells, which leads to immediate immunomodulation [101,102]. Hormone peptides secreted by EECs communicate with afferent VN fibers to release Ach. This inhibits inflammatory responses from surrounding immune cells. Vagal afferent signaling also alters fat deposits, leading to changes in the level of fat-secreted adipokines, such as leptin. This affects the functioning of CD4+ T-cells, with a direct influence on the functioning of peptide hormones [98].

Serotonin (5-HT) produced by enterochromaffin endocrine cells is recognized by seven receptor isoforms expressed on mast cells, monocytes, dendritic cells (DCs), eosinophils, T- and B-cells, and neutrophils [103]. Immune cells also produce 5-HT independently of endocrine cells [103]. Peptide hormone carboxypeptidase increased IL-6 and chemokine (C-X-C motif) ligand (CXCL) 1 in mice and led to intensified dextran sulfate sodium (DSS)-induced colitis [104]. CCK octapeptide has been shown to inhibit TLR9 stimulation of plasmacytoid DCs via tumor-necrosis-factor-receptor-associated factor 6 signaling [105], whereas it can promote IL-12 production from DCs and reduce IL-6 and IL-23 production, offering protection during collagen-induced arthritis [106]. CCK has a direct effect on T- and B-cells and led to the formation of a Th2 and regulatory T-cell phenotype in vitro [107]. In other studies, CCK increased IL-2 production in Jurkat T-cells [108], stimulated B-cells to produce acetylcholine [109], and reduced B-cell lipopolysaccharide (LPS)-induced activation [107].

Ghrelin increased T-cell proliferation via phosphoinositide 3-kinase, extracellular-signal-regulated kinases, and protein kinase C [110]. Ghrelin had an anti-inflammatory effect on DSS-induced colitis [38] and proved to have antiparasitic [111] and antibacterial effects [112]. Of interest is that ghrelin is also produced by T-cells [113].

EECs produce the pro-inflammatory cytokine IL-17C during CD and UC [114]. CCK-secreting EECs expressed TLR 1, 2, and 4, and led to an increase in NFκβ production, MAPK signaling, and Ca^2+^ flux in tumor necrosis factor-α, and the transformation of growth factor-β and macrophage inhibitory protein 2 [115]. In in vitro studies, EECs reacted differently to pathogens and nutrients by secreting CXCL1/3 and IL-32 in response to flagellin and LPS, but not to fatty acids [116]. EECs can thus act as front-line pathogen detectors and release either classical cytokines or peptide hormones. In other words, EECs can regulate adaptive and innate immunity. This is important in keeping the gut microbiome in a homeostatic state. If unbalanced, the gut enters a state of dysbiosis, characterized by a drastic increase in Enterobacteriaceae, especially *Escherichia*, *Shigella*, *Proteus*, and *Klebsiella*, and an increase in enterotoxin levels [60]. If left untreated, major gastrointestinal disorders, such as diarrhea, UC, CD, and other inflammable bowel diseases (IBDs), may develop [57,58,59]. In severe cases, elevated toxin levels may alter the functioning of the intestinal barrier and blood–brain barrier (BBB), which may lead to neurodegeneration [60,61,62,63].

## 5. The Role of Gut Bacteria in Mental Health

Several research studies have shown that dysbiosis is a contributing factor to the development of neurological and psychiatric diseases. These include anxiety, depression, major depressive disorder (MDD), schizophrenia, bipolar disorder, autism, and obsessive-compulsive disorder (OCD) [117]. Acetate, lactate, butyrate, and propionate produced by anaerobic bacteria in the large intestine have a profound effect on reactions causing mental health issues. Experiments on animals have shown that butyrate inhibits histone deacetylase (HDAC) [56,118]. Overproduction of HDAC is associated with neurological disorders such as Parkinson’s disease, schizophrenia, and depression [118]. The inhibition of HDAC may thus be important in the treatment of brain trauma and dementia [118]. An increase in acetylated histones (ACHs), on the other hand, elevates the expression of the *bdnf* (brain-derived neurotrophic factor) gene in the frontal cortex and hippocampus and stimulates brain development [119,120]. Low levels of BDNF are associated with mood changes, depression, and anxiety [94,121,122,123]. Activation of G-protein-coupled receptors (GPCRs) by butyrate may lead to the development of multiple neurodegenerative disorders [56,118] and an increase in the production of inflammatory cytokines [124]. Butyrate (and other SCFAs) modifies the integrity of the blood–brain barrier (BBB) and the maturation of microglia [125,126]. In germ-free (GF) mice, the malfunctioning of microglia could be reversed by administering high levels of a combination of butyrate, propionate, and acetate [127]. The function of acetate is different in that it crosses the BBB and accumulates in the hypothalamus, where it controls appetite [79].

*Oscillobacter* spp. are known to produce valeric acid, a compound that closely resembles GABA and binds to GABA receptors [128]. With the binding of valeric acid to these receptors, GABA binding is inhibited, and the CNS signals are no longer blocked, resulting in anxiety. Of interest is that patients who suffer from anxiety and depression have lower *Lactobacillus* cell numbers. This is an important observation as certain *Lactobacillus* spp. stimulates the secretion of GABA as well as the neurotransmitter acetylcholine [129,130].

Hormone production regulated by gut microbiota interacts with EECs [131,132] and generates chemical signals that, in turn, react with the ENS. Bodily functions affected by changes in hormone levels include digestion, salivation, lacrimation, urination, defecation, and sexual arousal [133]. A clear association exists between chronic stress and gut inflammation disorders, such as IBD and IBS [115].

Inflammatory factors produced by *Alistipes* spp. could play a role in depression, anxiety, IBD, and chronic fatigue [134,135,136]. *Prevotella* spp. are often associated with pro-inflammatory responses characteristics, but low numbers of the species have also been associated with anxiety and depression [137]. *Faecalibacterium prausnitzii* (ATCC 27766) relieved anxiety and depression, suggesting that the strain may have psychobiotic properties [138]. This may have to do with increased levels of SCFA in the cecum, elevated plasma IL-10 levels, and lower levels of corticosterone and IL-6 [139]. *Ruminococcus flavefaciens* upregulated genes involved in mitochondrial oxidative phosphorylation whilst downregulating genes involved in synaptic signaling and neurogenesis [140]. Significantly lower levels of *Proteobacteria*, *Haemophilus*, *Sutterella*, and *Clostridium* spp. were reported in schizophrenic patients, whilst cell numbers of *Anaerococcus* spp. remained unchanged. Not all studies came to the same conclusion. In another study [141], high levels of *Proteobacteria*, *Succinivibrio*, *Collinsella*, *Clostridium*, and *Klebsiella* spp. but low levels of *Blautia*, *Coprococcus*, and *Roseburia* spp. were reported in schizophrenic patients [142,143]. Concluded from the study conducted by Qing et al. [144], *Firmicutes* live in synergy with actinobacteria, fusobacteria, and Acidobacteria, especially during the early stages of schizophrenia. High cell numbers of *Firmicutes* were reported to be present in the saliva of patients with primary Sjögren’s syndrome [145]. High levels of lactic acid bacteria, especially *Lactobacillus gasseri*, were reported in schizophrenic patients. Bacteriodetes and *Acinetobacteria*, on the other hand, were in the minority [146]. In contrast to previous studies, the presence of *Proteobacteria* did not differ significantly between schizophrenic and non-schizophrenic patients.

*Anaerococcus* and *Collinsella* isolated from schizophrenic individuals are known to produce high levels of butyrate [142,143]. In bipolar and autistic patients, butyrate-producing *Faecalibacterium* spp. was present in low numbers [147]. Patients with bipolar disorder are often diagnosed with an increase in gut wall permeability [148] and an increase in *Flavonifractor* [149]. Species in this genus cleave the flavonoid quercetin [149]. The latter has antioxidative and anti-inflammatory properties [150], and changes in its levels could lead to bipolar disorder. Other changes in bacterial populations included a decrease in *Faecalibacterium* and an increase in *Actinobacteria* and *Coriobacteriaceae* [151]. Evans et al. [152] reported a decrease in *Faecalibacterium* and Ruminococcaceae in bipolar patients. The decrease in *Faecalibacterium* resulted in a decline in anti-inflammatory reactions [153].

An increase in *Clostridium* spp. was reported in patients diagnosed with autism [154]. In a more detailed study on the complete microbiome of autistic patients, Finegold et al. [155] reported a significant increase in Bacteroidetes, *Acintobacterium,* and *Proteobacterium* spp. but a decline in *Firmicutes* in autistic patients. High cell numbers of *Clostridium defense*, *Clostridium hathewayi*, and *Clostridium orbiscindens* were isolated from autistic patients. *Faecalibacterium* and *Ruminococcus* spp. were less abundant, which is an important observation given the anti-inflammatory properties of these species.

Obsessive-compulsive disorder (OCD), diagnosed in 2.3% of the population (mostly men), with a predominance in men [156], was originally classified as an anxiety disorder, similar to autism. Little is known about the gut microbial composition of individuals with OCD. Turna et al. [157] reported low numbers of *Oscillospira*, *Odoribacter*, and *Anaerostipes* spp. in OCD patients. Odoribacter produces butyrate, and low cell numbers may lead to an increase in inflammation, which may be the onset of OCD [158]. Experiments on mice showed a decrease in OCD when treated with *Lactobacillus rhamnosus.* In humans, similar findings were reported with the administration of *Lactobacillus helveticus* [159,160]. Findings such as these may provide valuable information in future research aimed at developing psychobiotics. For more information on probiotics and the effect on the nervous system, the reader is referred to the review published by Cryan et al. [161].

The production of trace amines β-phenylethylamine (PEA), *p*-tyramine (TYR), and tryptamine (TRP) by commensal gut microbiota is well-documented [162,163,164] as many gut bacteria can produce aromatic L-amino acid decarboxylase (AADC; EC 4.1.1.28) [165]. Unlike DOPA, NE, EPI, and 5-HT, PEA, TYR, and TRP are not stored and rapidly diffuse across membranes [166,167]. PEA diffuses across the blood–brain barrier and TYR passes through IECs [168]. Tyrosine is converted to l-3,4-dihydroxyphenylalanine (l-DOPA), the precursor of DOPA, NE, and EPI [168]. A deficiency in L-tyrosine may thus lead to anxiety and low mood [168]. Treatments that increase monoamine neurotransmitter receptor activation lead to a decrease in PEA and TYR synthesis. Likewise, treatments that decrease receptor activation result in an increase in PEA and TYR synthesis. Reports on changes in AADC activity are almost exclusively based on L-DOPA as a substrate. Binding of PEA, TYR, TRP, and OCT to the trace-amine-associated receptor TAAR1 in the brain regulates the release of neurotransmitters dopamine and serotonin [169]. Under- or overexpression of TAAR1 may lead to schizophrenia, depression, and addiction [170].

The gut microbiota has a vast effect on our mental health. Chemicals secreted by these bacteria, such as GABA, in addition to other metabolites, play an important role in anti-inflammatory responses and help to alleviate psychiatric symptoms stemming from inflammation. Treatment of schizophrenic and bipolar patients with probiotics alleviated symptoms associated with IBD, autistic children benefitted from probiotic treatment, and OCD-like behavior could be controlled (reviewed by Dicks et al.) [117]. The same review addresses the role that specific gut microbiota play in mental health and lists the species responsible for each mental condition.

## 6. Conclusions

The gut microbiota has a vast impact on the GBA and our overall mental health, as shown by drastic changes in the production of neurotransmitters when signals from the CNS reach the ENS. These changes also affect digestion, intestinal permeability, gastric motility, and immune regulation. Many of the metabolic compounds produced by gut bacteria play an important role in anti-inflammatory responses and, because of this, also alleviate psychiatric disorders caused by inflammatory responses. Recent studies have shown that schizophrenia, bipolar disorder, autism, and OCD may be prevented, and perhaps treated, by maintaining a healthy, balanced gut microbiome. Much more research is required to understand how gut microorganisms control cognitive behavior, mood, and neuropsychiatric disorders. This entails deciphering the complex, everchanging network between cells and neurons. Further research is required to understand the interactions between gut microbiota. The identification of changes in the gut microbiome associated with psychological disorders may provide valuable information in the choice of treatment. Further in-depth studies need to be performed on the effect signals generated by the hypothalamus and pituitary and adrenal glands have on the ENS. This is important as the entero-endocrine signaling pathway and immunological reactions keep the gut microbiome in a balanced state.

## Figures and Tables

**Figure 1 ijms-25-00038-f001:**
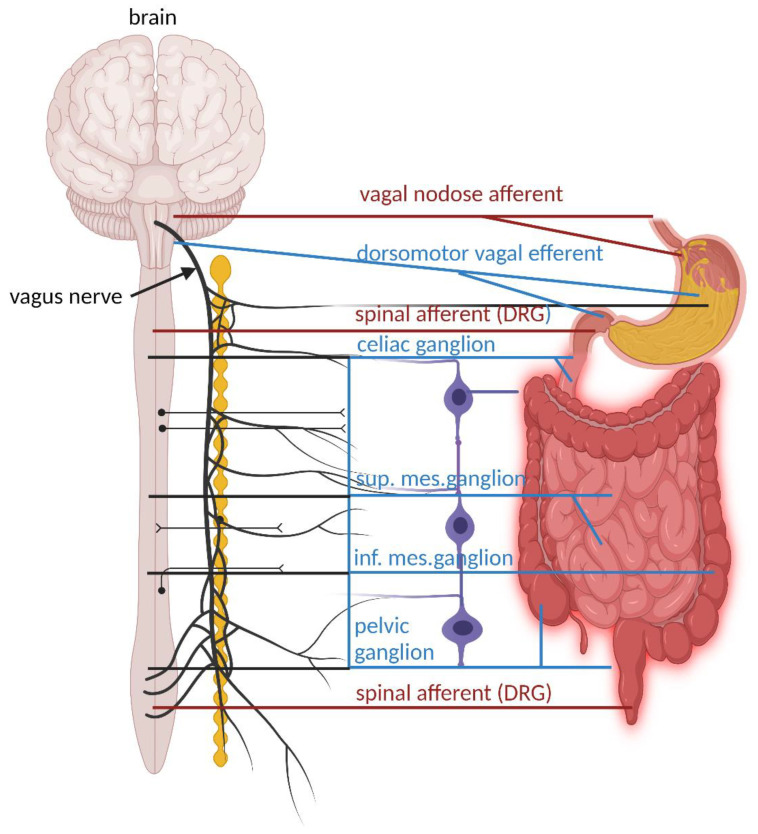
Major extrinsic neural pathways between the ENS, spinal cord, and the brain. Extrinsic motor pathways are shown with blue lines. These pathways include the parasympathetic (vagalmotor) and sympathetic nervous systems stemming from the thoracolumbar spinal cord and linking with the celiac, superior mesenteric ganglia, and inferior mesenteric ganglia (blue lines). In the esophagus and stomach, the major extrinsic sensory nerves arise from the vagus nerve (red line). In the colon, the major extrinsic sensory nerves stem from spinal afferent nerves (red line), with cell bodies in dorsal root ganglia (DRG). This illustration was constructed using BioRender (https://biorender.com/, assessed on 20 November 2023).

**Figure 2 ijms-25-00038-f002:**
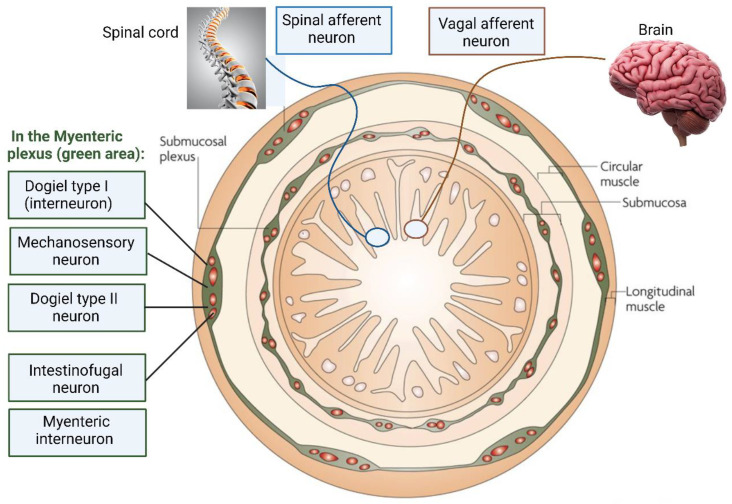
Intrinsic sensory neurons and extrinsic sensory nerve endings in the ENS. Dogiel type I neurons, located in the myenteric plexus, are interneurons and are length-sensitive but tension-insensitive. Mechanosensory neurons are rapidly adapting and excitatory. Dogiel type II neurons, also in the myenteric plexus, are chemosensory and mechanosensitive and react to rapid and slow synaptic inputs from other enteric neurons. Intestinofugal neurons are “second-order” neurons but react to direct mechanical stimuli. Extrinsic vagal afferent nerve endings supply signals to mostly the upper gut and act as slowly adapting tension receptors. Spinal afferent neurons send signals to the lower gut (distal colon) and are activated by changes in muscle tension. This illustration was constructed using BioRender (https://biorender.com/, assessed on 20 November 2023).

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
