# Peer review of "Our Mental Health Is Determined by an Intrinsic Interplay between the Central Nervous System, Enteric Nerves, and Gut Microbiota"

_ijms, 2023, doi:10.3390/ijms25010038_

Round 1

Reviewer 1 Report

Comments and Suggestions for Authors

Journal name: International Journal of Molecular Sciences

Title: Our Mental Health is Determined by an Intrinsic Interplay Between the Central Nervous System, Enteric Nerves, and Signals Generated by Gut Microbiota

Reviewer Report:

I read with interest this review article which aimed to discuss the relationship between the central nervous system, enteric nerves, and signals generated by gut microbiota. The submitted manuscript seems to be well-structured and organized. It includes all necessary parts. Nevertheless, from a formal and substantive point of view, I have some suggestions which should be considered:

1) The title should be moderately revised to be more attractive

2) As stated in the abstract, one of the main aims of this review is to discuss the complex control of enteroendocrine signaling and immune responses that maintain the gut microbiome in a state of equilibrium. This aspect should be clearly explored and mentioned as a title or subtitle or sub-section, showing the role of immune responses

3) The title of section 4. Mental health should be revised, I suggest mentioning the role of the gut microbiota in mental health. A paragraph demonstrating the impact of probiotics on the nervous system should be added.

4) The addition of tables summarizing the different players involved in our mental health, with the experimental methods used and the main molecular mechanisms involved, etc., would be very interesting.

Author Response

Dear Reviewer,

Thank you for the comments. All queries have been addressed and changes in the text are highlighted.

I read with interest this review article which aimed to discuss the relationship between the central nervous system, enteric nerves, and signals generated by gut microbiota. The submitted manuscript seems to be well-structured and organized. It includes all necessary parts. Nevertheless, from a formal and substantive point of view, I have some suggestions which should be considered:

  • The title should be moderately revised to be more attractive

Answer: The title has been slightly changed, as requested, by changing it from  “Our Mental Health is Determined by an Intrinsic Interplay Between the Central Nervous System, Enteric Nerves, and Signals Generated by Gut Microbiota” to “Our Mental Health is Determined by an Intrinsic Interplay Between the Central Nervous System, Enteric Nerves, and Gut Microbiota”. (The last part of the title has been shortened).

  • As stated in the abstract, one of the main aims of this review is to discuss the complex control of enteroendocrine signaling and immune responses that maintain the gut microbiome in a state of equilibrium. This aspect should be clearly explored and mentioned as a title or subtitle or sub-section, showing the role of immune responses

Answer: A new section, Section 4 (Enteroendocrine Signaling and Immune Response) has been added (lines 276 to 328).  The discussion is supported by the addition of 19 new references.

  • The title of section 4. Mental health should be revised, I suggest mentioning the role of the gut microbiota in mental health. A paragraph demonstrating the impact of probiotics on the nervous system should be added.

Answer: The title (now Section 5) was changed to “The Role of Gut Bacteria in Mental Health”.

I feel the impact of probiotics has been addressed throughout the review, e.g. lines 153-154, 188, 204, 205, 209, 210, 247, 354, 379, 380, 408, and 409.  In addition, two papers addressing the topic (reference 161, mentioned in lines 411 and 412, and reference 117 mentioned in lines 431 to 435), the latter being one of my previous papers, addressed the topic of probiotics substantially.

  • The addition of tables summarizing the different players involved in our mental health, with the experimental methods used and the main molecular mechanisms involved, etc., would be very interesting.

Answer: A paragraph has been added at the end of Section 5 (lines 428-435), with reference to a table (list of strains) included in one of my published papers (reference 117) under the heading “Role of Gut Microbiota in Psychiatric Disorders”. This should provide the reader with sufficient information.

Yours sincerely

Prof LMT Dicks

Reviewer 2 Report

Comments and Suggestions for Authors

The current review focuses on the interaction processes and mediators in the gut-brain axis, with a specific emphasis on mental health. While the chosen topic of the paper has the potential to engage readers, there are notable shortcomings in content coverage and organizational logic, leading to the omission of crucial information.

  1. The author lists many contents that have already been summarized and reported in existing reviews. The paper lacks innovative viewpoints and ideas, primarily summarizing previous research without presenting novel contributions.

  2. The paper mentions two main interaction pathways, involving the nervous system and signals generated by gut microbiota. However, the central and intestinal immune systems, crucial interaction pathways, are not thoroughly explored in this paper.

  3. Building on existing literature, specific interaction pathways should be subdivided. For instance, while the author emphasizes the role of microbiota, besides microbial regulation of the nervous system through metabolism, there are various other ways microbes can influence, which the author fails to address and should be supplemented.

  4. In the section on "The influence of gut bacteria on the ENS," it is recommended to supplement a table detailing key microbiota to facilitate reader comprehension.

In conclusion, while the paper's theme has the potential to capture reader interest, the overall organization and logic of the content are lacking. The author does not exhibit an in-depth grasp of the latest research in the relevant field. Therefore, a major revision is suggested, and upon addressing these concerns, the manuscript can be reconsidered for acceptance or not.

Comments on the Quality of English Language

No

Author Response

Dear reviewer

Your comments are much appreciated, thank you. I have addressed these below and changes in the text are highlighted.

The current review focuses on the interaction processes and mediators in the gut-brain axis, with a specific emphasis on mental health. While the chosen topic of the paper has the potential to engage readers, there are notable shortcomings in content coverage and organizational logic, leading to the omission of crucial information.

The author lists many contents that have already been summarized and reported in existing reviews. The paper lacks innovative viewpoints and ideas, primarily summarizing previous research without presenting novel contributions.

Answer to the above two points: The purpose of the review is to summarize the interaction processes and mediators in the gut-brain axis, with a specific emphasis on mental health.  An additional section (Section 4) has now been added.  Topics and areas that need to be addressed in future research are mentioned in the Conclusion section.

The paper mentions two main interaction pathways, involving the nervous system and signals generated by gut microbiota. However, the central and intestinal immune systems, crucial interaction pathways, are not thoroughly explored in this paper.

Answer: As stated above, a new section, Section 4 (Enteroendocrine Signaling and Immune Response) has been added (lines 276 to 328).  The discussion is supported by the addition of 19 new references.

Building on existing literature, specific interaction pathways should be subdivided. For instance, while the author emphasizes the role of microbiota, besides microbial regulation of the nervous system through metabolism, there are various other ways microbes can influence, which the author fails to address and should be supplemented.

Answer: This has now been addressed with the additional information added to the review (lines 276-328).

In the section on "The influence of gut bacteria on the ENS," it is recommended to supplement a table detailing key microbiota to facilitate reader comprehension.

Answer: A paragraph has been added at the end of Section 5 (lines 428-435), with reference to a table (list of strains) included in one of my published papers (reference 117) under the heading “Role of Gut Microbiota in Psychiatric Disorders”. This should provide the reader with sufficient information.

In conclusion, while the paper's theme has the potential to capture reader interest, the overall organization and logic of the content are lacking. The author does not exhibit an in-depth grasp of the latest research in the relevant field. Therefore, a major revision is suggested, and upon addressing these concerns, the manuscript can be reconsidered for acceptance or not.

Answer: I respectfully disagree with this statement.  Previous papers have been published in this field (Dicks, L.M.T. Gut bacteria and neurotransmitters. Microorganisms 2022, 10, 1838; Dicks, L.M.T. How does quorum sensing of intestinal bacteria affect our health and mental status? Microorganisms 2022, 10, 1969; and Dicks, L.M.T.; Hurn, D.; Hermanus. Gut bacteria and neuropsychiatric disorders. Microorganisms 2021, 9, 2583).  The current review discusses the relationship between the central nervous system, enteric nerves, and signals generated by gut microbiota.  I feel the manuscript is well-structured and organized, as also pointed out by one of the other reviewers.  The addition of Section 4 provides more in-depth information.

Yours sincerely

Prof LMT Dicks

Round 2

Reviewer 1 Report

Comments and Suggestions for Authors

Accept in present form

Reviewer 2 Report

Comments and Suggestions for Authors

Accepted.